# A Review on Chemical Vapour Deposition of Two-Dimensional MoS_2_ Flakes

**DOI:** 10.3390/ma14247590

**Published:** 2021-12-10

**Authors:** Luca Seravalli, Matteo Bosi

**Affiliations:** IMEM-CNR, Parco Area delle Scienze 37A, 43124 Parma, Italy

**Keywords:** chemical vapour deposition, 2D materials, MoS_2_

## Abstract

Two-dimensional (2D) materials such as graphene, transition metal dichalcogenides, and boron nitride have recently emerged as promising candidates for novel applications in sensing and for new electronic and photonic devices. Their exceptional mechanical, electronic, optical, and transport properties show peculiar differences from those of their bulk counterparts and may allow for future radical innovation breakthroughs in different applications. Control and reproducibility of synthesis are two essential, key factors required to drive the development of 2D materials, because their industrial application is directly linked to the development of a high-throughput and reliable technique to obtain 2D layers of different materials on large area substrates. Among various methods, chemical vapour deposition is considered an excellent candidate for this goal thanks to its simplicity, widespread use, and compatibility with other processes used to deposit other semiconductors. In this review, we explore the chemical vapour deposition of MoS_2_, considered one of the most promising and successful transition metal dichalcogenides. We summarize the basics of the synthesis procedure, discussing in depth: (i) the different substrates used for its deposition, (ii) precursors (solid, liquid, gaseous) available, and (iii) different types of promoters that favour the growth of two-dimensional layers. We also present a comprehensive analysis of the status of the research on the growth mechanisms of the flakes.

## 1. Introduction

The information technology revolution is driven by the continuous improvement of electronic devices and their constant scaling down. However, this process is rapidly approaching its limit. In order to continue the innovation process, introduction of new concepts and new materials is required. Some solutions have been identified in (i) the use of strain engineering and high-k gate dielectrics, (ii) silicon–germanium (SiGe) alloys and germanium, because of their higher electron mobility and lower need for power, and (iii) nanostructures and 2D materials, for the possibility of introducing brand-new device designs and concepts.

For the latter, graphene and similar 2D materials recently emerged as promising candidates because they exhibit interesting mechanical, electronic, optical, and transport properties with peculiar differences from those of their bulk counterparts. Graphene, despite its exceptional physical properties, has the major drawback of lacking an electronic bandgap, a limit that poses problems in realizing logic circuits and transistors. Bandgap engineering in graphene, although possible, has several drawbacks, such as increased complexity of the process and degradation of material quality, so it is not usually considered a viable choice for certain applications.

Transition metal dichalcogenides (TMD), silicon, germanium, and boron nitride can be thinned down to monolayers similarly to graphene, exhibiting weak interplane interaction and strong in-plane bonds. MoS_2_ is one of the most promising materials of this family because of the relative easiness of its synthesis and its interesting physical properties. It exhibits a hexagonal lattice structure consisting of a layer of transition metal atoms embedded in two chalcogen layers, with strong covalent bonds in each 2D plane and a weaker interaction between different planes. The properties of MoS_2_, similarly to those of other 2D materials, depend mainly on the vertical thickness of MoS_2_ rather than its lateral size. The most important difference with respect to graphene is that 2D MoS_2_ shows a direct bandgap of about 1.8 eV, making it an ideal candidate for applications in electronics, photonics, photovoltaics, energy storage, and catalysis [1].

Several proof-of-concept MoS_2_ devices have been already demonstrated. FETs with mobility up to 320 cm^2^ V^−1^ s^−1^ and on/off ratio of 10^8^ at room temperature were realized [2,3], as well as inverters with gain up to 16 [4]. Several kinds of phototransistors, photosensors [5], and gas and biological sensors [6] were demonstrated, and the mechanical properties of MoS_2_ were exploited in the realization of strain sensors [7]. Moreover, the light emission properties of this material are also attracting considerable interest [8,9,10,11].

The future development of research is trending towards the integration of different building blocks and different 2D materials into a single device, aiming for the fabrication of high-performance CMOS-based circuits, sensors, and efficient photocatalytic systems [12,13].

The interest in developing new devices from 2D structures is also driven by several novel properties observed in these materials, such as valley polarization, in which conduction and valence bands of MoS_2_ monolayers present two inequivalent valleys at points K and K_0_, giving a new degree of freedom to carriers. This would permit radical new device concepts based on valleytronics and on spin–valley coupling, permitting control of information using circular polarized light [14].

Despite the results obtained so far on MoS_2_ devices and transistors in particular, it was observed that their performance was still far from the best theoretical predictions. The transport properties of MoS_2_ are limited by defects such as vacancies, antisites, charge traps, grain boundaries, and Coulomb impurities at the flake interface [15,16]. For example, the maximum predicted electron mobility of MoS_2_ at room temperature is about 410 cm^2^ V^−1^ s^−1^, leaving room for improvement. Understanding these limits and how to overcome them is a major issue on the MoS_2_ roadmap.

Reliable and controlled doping in MoS_2_ flakes is also a fundamental requirement to achieve precise control of their electrical and optical properties. Usually, substitutional doping is the dominating process as compared to interstitial doping. On this topic, fundamental contributions were the very recent reviews of Lin et al. [17] and Rai et al. [18], who considered the issues of Schottky barriers and contact resistance at the interface between MoS_2_ and a metal and the problems related to the doping of a wide range of 2D TMD materials, respectively. N-type doping of MoS_2_ could be obtained by adding ReO_3_ to the solid precursors [19] or by using ZnS as a sulfur precursor [20], while chloride permitted decreasing the resistivity of MoS_2_ flakes from some KΩ to about 0.5 KΩ, allowing for the realization of Schottky barriers with enhanced properties [21]. p-doping could be obtained by adding Nb_2_O_5_ to NaCl promoter when using solid precursors [22].

Another topic of particular interest is the possibility of including elements such as manganese (Mn) and iron (Fe), aiming to add new functionalities related to magnetism and spin to MoS_2_. First reports indicated the possibility of incorporating Mn in MoS_2_, but with some issues relative to the substrate used, as Mn was detected only in structures deposited on graphene and not on those deposited on SiO_2_/Si [23].

Despite their peculiar properties and the achievements so far obtained, the controllable synthesis of large-area flakes and the reproducibility of the process are still challenging issues; these are two essential, key factors required to sustain the future development of 2D materials. Mechanical exfoliation has so far granted a very high material quality, leading to the realization of devices and to an in-depth study of their properties. However, this method is very impractical and poses many limits on large-scale production. Industrial application of TMD is directly linked to the development of a high-throughput and reliable technique to obtain 2D layers of different materials on large area substrates, with a simple, reproducible, and scalable method. Chemical vapour deposition (CVD) is a good candidate for this task, since it has already been widely adopted for the mass production of many kinds of devices based on III-V and III-N materials and has thus the potential to permit the integration of 2D materials with “standard” semiconductors or other compounds. Many excellent reviews on MoS_2_ are already available, giving a broad scenario of all the work done on this material [24,25].

Although MoS_2_ flakes with large size (>500 μm) have been obtained with relevant success from several groups [26,27,28], in our opinion, research is still needed on many points that are usually overlooked: (i) the effect of different parameters (such as carriers’ flows or sulfur partial pressure), (ii) the possibility of engineering layers by surface treatments—this is crucial for the realization of heterostructures and possibly hybrid nanostructures, (iii) the growth reproducibility of these structures. It is a common experience to not be able to obtain the same 2D layers in different growth runs, and this is still a consistent problem for the technology’s transfer towards industrial applications. A reliable growth protocol that can be transferred to different reactors with minimal setup time is a strong requirement to allow 2D MoS_2_ to make the jump to mass production.

The preparation of this material is still challenging, and there is a lot of room for improving both the process and the material quality. Despite its relative simplicity, CVD remains one of the most widely used techniques to synthesize TMD flakes. Recently, new protocols and precursors have been introduced to optimize the deposition, to obtain better quality material, and to improve the reproducibility of the process.

The aim of this work was to collect and critically present the knowledge acquired so far in this field, providing the reader with a comprehensive review of the state of the art of the CVD growth processes used to prepare MoS_2_.

## 2. CVD Setup for Growth of MoS_2_

The basic concept of CVD is very simple, and it is schematically presented in Figure 1a: the precursors—either only S or both S and Mo compounds—are delivered in gaseous form to a substrate (placed on a susceptor of graphite or similar material) kept at high temperature, where the chemical reactions needed for the deposition of MoS_2_ occur. Most of the results reported in literature have been obtained using a standard horizontal reactor, such as that sketched in Figure 1. This configuration, although widely used, has the drawback of a nonuniform precursor gradient along the flow direction, resulting in a nonuniform deposition and in intrinsic difficulty in optimizing the growth parameter, since small changes in substrate position or in precursor supply may cause nonreproducible results. Development of vertical CVD reactors may help in solving these issues, since with this configuration, temperature and precursor supply may be more easily controlled and homogenized [29].

In the standard horizontal configuration, a quartz tube about 1 m long with a diameter of 2–5 cm is used. The heating is provided by a resistance placed around the tube. The reaction for MoS_2_ deposition requires a temperature in the 600–800 °C range, but in some cases, the supply of sulfur requires a second heating zone in the 100–200 °C range. In this case, to obtain better reproducibility and control, it is more convenient to separately and independently heat this additional zone (T2 in Figure 1a) in which the sulfur precursor is placed.

It is also possible to deposit the molybdenum precursors directly on the substrate, either in solid form (Figure 1b) or as a liquid solution (Figure 1c). Growth promoters, to obtain better control of the process, can be added either directly on the growth substrate or on a different substrate (Figure 1d). As a last method, the use of gaseous precursors (Figure 1e) avoids the need for placing the precursors inside the tube before the beginning of the process.

A standard routine to minimize the contamination from external air and O_2_ in the reaction tube is to purge the system with inert gas several times before starting the growth, eventually evacuating the tube with a rotary pump and then flushing with purified Ar or N_2_ up to ambient pressure, repeating this procedure several times.

The actual CVD process depends on the chemical status of the used precursors (solid, liquid, gaseous), but it is nevertheless possible to divide it into the following steps:

The precursors are brought into gaseous form and diluted in an inert carrier gas; e.g., powders are evaporated/sublimated, or a controlled amount of gas is measured by means of a mass flow controller.

The reactive species are transported by the carrier gas to the substrate. Chemical reactions may also occur in this step, e.g., reducing reactions.

The precursors diffuse towards the substrate surface.

The precursors are adsorbed at the surface, where adatom adsorption and migration occurs. The MoS_2_ flakes synthesis occurs in this step. By-products re-evaporate or desorb into the gas streams and are carried away into the exhaust.

The heating ramp times are usually in the range of 10–30 min from room temperature to growth temperature. The heating of sulfur powders typically starts when the substrate is already at the target temperature; this permits avoiding injecting S into the system before a constant substrate temperature is reached, limiting prereactions. The MoS_2_ growth proceeds for about 10–20 min, and then the system naturally cool downs to ambient temperature. If powders are used as reagents, heating of the substrate and of the MoO_3_ powders occurs simultaneously, since they are usually placed very near the substrate. This means that Mo starts to evaporate before the substrate reaches the target temperature.

Flake synthesis is dependent on different parameters such as the growth temperature and the distance between the substrate and the powders. Flow rate, chamber pressure, precursor supply, powder dimension, and purity also affect the final outcome of the growth. Despite the simplicity of the process, if excellent control is not achieved, the results may not be completely reproducible, and large-scale deposition can be very hard to obtain.

While it has been shown that different carrier gases can strongly influence the growth of graphene [30], considerably less attention has been devoted to this topic for the growth of MoS_2_. Considering that S is a strong reducing agent, the use of a purified inert gas (N_2_ or Ar) is usually sufficient for the sulfurization reaction to occur, and the carrier gases can be considered as inert in the reactions at the basis of 2D nucleation and growth. The use of H_2_ is thus not common in the MoS_2_ CVD process. Nevertheless, it was argued that H_2_ can have beneficial effects, as it can inhibit the thermally induced etching effect and promote the desulfurization reaction [31]. By carefully adjusting the amount of hydrogen in a H–Ar mixed carrier gas, authors were able to obtain layers with high crystallinity and a nearly perfect S/Mo atomic ratio. It was also suggested that, in the case of MoS_2_ growth using Mo(CO)_6_ with (C_2_H_5_)_2_S as precursors, the presence of H_2_ is necessary for removing carbonaceous species generated during the MOCVD growth, permitting increasing the average grain size from hundreds of nanometres to more than 10 μm [32].

Carrier gas flow rate was reported to affect the morphology of flakes, as well as their size and density: lowering the gas carrier flow during the heating stage from 35 to 15 sccm permitted increasing the density and size of the flakes [33]. Moreover, the shape of MoS_2_ flakes changed from zigzag to triangular as the gas carrier flow was decreased, indicating higher material quality. This behaviour was directly related to the quantity of S in the gas phase. As the carrier flow increased, the S transport grew more efficient, and S reacted with MoO_3_ powders, suppressing their evaporation, lowering MoO_3_ partial pressure, and promoting zigzag edge termination.

In order to have better control over S and MoO_3_ reactions during the heating stage of the process, a procedure of “flow reversal” was proposed [34]. During the heating of the furnace, Ar was introduced from the side of the Mo powders towards the side of the S powders (Figure 2). In this way, S reached neither the substrate nor the Mo powders, and any unintentional reaction was prevented; growth and nucleation of MoS_2_ far from the steady growth regime was inhibited. Once the setpoint temperatures of both substrate and S powders were reached, the Ar flow direction was switched back to the standard direction, so that it entered from the S powder side and delivered S vapours to the Mo powders and the substrate. Using this two-stage growth, the mean side length of MoS_2_ flakes was increased up to 250–300 μm.

Another custom setup to obtain better control of the precursor supply and to avoid cross-contamination of MoO_3_ and S powders consisted of placing S powders in a small tube nested inside another, larger tube so that S did not contaminate the MoO_3_ powders before entering the reaction zone [35].

## 3. Substrates

A typical characteristic of 2D materials is the very weak interlayer Van der Waals forces, in contrast with the stronger in-layer chemical bonds, which are ionic or covalent. This is the reason why bulk crystals of TMD materials, and MoS_2_ in particular, can be easily cleaved and exfoliated to produce few-layer flakes. As a consequence, the dangling bonds of the 2D plane are relatively inactive, and few interactions occur out-of-plane. For this reason, one could argue that in the 2D synthesis, the substrate’s role should be less important with respect to standard epitaxy, in which lattice and thermal matching conditions between the substrate and the epilayer are of paramount importance, determining whether defects are generated at the interface. The weak interactions between the substrate and the 2D layer reduce the importance of lattice mismatch, but the choice of the substrate remains an important parameter, not to be overlooked. The reason should be sought in the processes still occurring at the substrate surface that involve the reactive species, such as adsorption, diffusion, and nucleation. It was indeed observed that the substrate played an important role in influencing both the growth process and the properties of 2D layers, as evidenced in the excellent review on this topic in [36].

SiO_2_/Si substrate is the most commonly used for CVD synthesis of MoS_2_ because of: (i) the ease of observing and characterizing flakes on it through optical microscopy (from this point of view, the thickness of the oxide layer is of paramount importance [37]); (ii) its high melting point; (iii) its compatibility with silicon processing. On the other hand, for further characterization and/or applications, it is necessary to transfer the 2D material from the SiO_2_/Si substrate on other substrates or TEM grids. This implies a delicate chemical process composed of various steps: (i) coating of the material with poly(methyl methacrylate) (PMMA) film on the MoS_2_ layers by spin coating, (ii) etching of the SiO_2_ layer to separate it from the Si substrate in a solution (KOH, HF, NaOH), (iii) transfer of the floating PMMA/MoS_2_ film from the surface of the solution to the substrate of interest, and (iv) removal of the PMMA coating by rinsing in acetone and isopropanol [38].

The possibility of growing MoS_2_ on rigid metals has been investigated, in particular for the case of gold, as it was demonstrated that excellent nanostructures can be obtained on Au foils [39]. If one considers surfaces of Au(111), very large nanoflakes can be obtained [40] thanks to the lattice matching between Au(111) and MoS_2_ and the fact that the step edge of Au(111) favours the unidirectional nucleation of the film. The advantage of this substrate was confirmed recently in [41], where a unidirectional growth ratio over 99% of MoS_2_ flakes was achieved on Au(111) films sputtered on c-sapphire. The growth on Ag(111) was also demonstrated, opening up interesting perspectives on the realization of metal contacts on this 2D material [42].

Liquid metals are interesting, as their uniform and smooth surface can inhibit the inhomogeneous nucleation of 2D materials. Although convincing results have been published on other 2D materials, the growth of MoS_2_ on these substrates has been reported only for the MoS_2_/h-BN heterojunction grown on liquid metal Ni–Ga alloy [43].

MoS_2_ has also been grown with considerable success on c-sapphire, as it has a stable hexagon single crystal structure, matching the lattice symmetry of 2D material [27,44,45]. Sapphire surface can be prepared with a thermal treatment at 1100 °C before MoS_2_ growth to obtain a clean and atomically flat surface. Scaling up to a Metal–Organic Vapour Phase Epitaxy (MOVPE) process for 6-inch wafers was obtained, and the possibility of transferring the obtained flakes to a secondary carrier substrate opens up the possibility to reuse the sapphire substrate, making this choice an interesting alternative to the common SiO_2_/Si approach [46]. Very recently, the growth of monolayer flakes as large as 50 μm on a C-plane sapphire was demonstrated by designing the miscut orientation towards the A axis. This resulted in a break in the degeneracy of nucleation energy for the antiparallel MoS_2_ domains [47].

The growth of a 6-inch uniform monolayer of MoS_2_ on solid soda-lime glass was achieved because of the advantageous effect of sodium in enhancing the formation of flakes that were homogenously distributed in glass [48]. Na acts as a growth promoter; its effects are discussed in depth in a subsequent section

Recently, the possibility of growing MoS_2_ on flexible substrates such as polyimide was reported [49]. The issues related to the stability of the substrate were solved thanks to a CVD growth protocol allowing for low temperatures (<300 °C), and the feasibility of developing flexible gas sensors taking advantage of the properties of MoS_2_ was demonstrated.

In an interesting work, MoS_2_ monolayers were reproducibly deposited on a variety of substrates, amorphous (SiO_2_/Si and fused quartz), crystalline (bare Si and sapphire), and layered-flexible (mica), using the same growth conditions with solid precursors and mixed NaCl [50].

CVD growth of MoS_2_ on gallium nitride (GaN) was also reported, a very relevant step towards the realization of heterostructures for many applications [51]. This possibility is of particular interest because it would eliminate the delicate step of detachment and transfer of 2D flakes, allowing realizing MoS_2_ heterojunctions directly on a nitride template.

Following the ideas widely explored with III-N and III-V semiconductors, in which AlGaAs–InGaAs or AlGaN–InGaN heterostructures are used to exploit the possibility of bandgap engineering to design advanced devices, an emerging topic of research is the realization of so-called 2D Van der Waals heterostructures. By vertically stacking layers of different 2D materials coupled by very weak forces, it would be possible to avoid the presence of defects due to lattice mismatch observed in conventional 3D heterostructures [52]. A considerable effort has thus been devoted to exploring the synthesis of MoS_2_ and other 2D compounds onto graphene and other 2D materials. First results highlighted how it was possible to obtain high-quality 2D flakes by adding some gases to gas carriers (N or Ar) for the treatment of graphene surfaces, such as H_2_ [53] or ozone [54]. This allows for the preparation of high-quality nanoflakes of MoS_2_ because of the reduction in the oxidation of graphene during the growth process.

In [55,56], few-layer MoS_2_ was grown on graphene oxide films or flakes. These results suggested that carbon-based materials can significantly promote the growth rate and yield of MoS_2_. In [57], the differences in using graphene, sapphire, and SiO_2_/Si substrates were investigated, highlighting the role of surface diffusion mechanisms in determining different properties of MoS_2_ flakes. The authors remarked that the growth on graphene was very stable, resulting in the realization of a strain-free 2D layer.

A topic of paramount importance for the development of nanodevices is the growth on patterned substrates to increase the spatial control of MoS_2_ nanoflakes and to obtain a controlled nucleation only on certain substrate zones. The first pioneering work on this issue was the one by Najmaei et al. [58], who observed a catalytic effect of the edges of the substrate. Based on this effect, authors proposed and demonstrated a method to control the growth of 2D flakes using lithography to pattern the substrate. Later, it was demonstrated that, by using patterned seeds of molybdenum, it was possible to obtain flakes of MoS_2_ at predetermined locations with a good spatial resolution [59]. Another approach to control the nucleation site of MoS_2_ flakes was to use droplets of (NH_4_)_2_MoS_4_ in dimethyl-formamide suspended from the tip of a micropipette that is dragged across a sapphire single-crystalline substrate by a controlled substrate movement. This resulted in MoS_2_ films with alternating mono- and few-layer regions that had distinct optical properties [60].

Other metal seeds proven as very valuable for the spatial control of the nucleation of flakes include Pt/Ti and Au [61,62,63]. More recently, the synthesis of patterned MoS_2_ nanoflakes using an industrial inkjet fed with an aqueous solution of ammonium molybdate tetrahydrate as liquid precursor was demonstrated (Figure 3). The possibility of obtaining large-area patterned 2D films in centimetre size with good controllability of the thickness and good reproducibility opens up interesting possibilities for the development of complex devices based on this material [64].

## 4. Precursors for MoS_2_ Growth

The literature has reported the use of several kinds of Mo and S precursors for the CVD growth of MoS_2_. It is possible to divide these precursors into three main categories according to their physical status: solid, liquid, and gaseous. To optimize the synthesis process and to obtain good control over the number of layers, it is important to understand the limits, advantages, and chemical and physical properties of each precursor. Moreover, for some precursors, several safety issues should be considered. In the following paragraphs, a brief description of the available precursors for each category is given.

## 5. Solid Precursors

Mo and S powders are still the precursors of choice for many groups to deposit mono- and few-layer MoS_2_, although the reproducibility of the process remains an issue. This approach is very simple, since the handling of powders is easy, has few hazard problems, and nevertheless permits obtaining high quality flakes. The drawback is that the flake dimension is somewhat limited to some tens of microns; scalability to large areas and enlargement of the single flake are quite difficult, and reproducibility remains an issue.

### 5.1. Molybdenum

The most commonly used powders for molybdenum are MoO_3_, available in high purity (99.98%). The vapour pressure of MoO_3_ was reported in [65]. A quantity of 10–200 mg of MoO_3_ powders is placed in a quartz crucible near the substrate in the high temperature zone of the reactor. There are several approaches in positioning the substrate; some groups place the substrate facing down directly above the powders, while others place it facing up immediately after the crucible. It should be noted that some impurities might be present in these solid precursors. Robertson et al. [66] showed that Cr and V atoms were found in CVD grown MoS_2_, most probably from elements present in solid precursors, causing some charge trapping in the grown 2D material. The issue of unintentional doping is very relevant in 2D materials; in [67], the effect of unintentional carbon doping during CVD was discovered and thoroughly discussed.

Precursors such as molybdenum hexacarbonyl (Mo(CO)_6_) and molybdenum (V) chloride (MoCl_5_) are mostly used for ALD and MOVPE. Mo(CO)_6_ is used with H_2_O or ozone to deposit thin MoO_3_ films through ALD, to be sulfurized in a subsequent step. Mo(CO)_6_ has a vapour pressure of about 0.10–0.15 mbar at room temperature and is very volatile (melting point of 150 °C, boiling point of 156 °C) [68].

Molybdenum (V) chloride (MoCl_5_) powders were also used to grow MoS_2_ with S powders [69,70]. However, MoCl_5_ is air sensitive and toxic, so its use poses more hazards than that of MoO_3_ powders.

Deposition of a thin Mo layer prior to growth could also be included in the solid precursor category. Electron-beam evaporation or sputtering are used to obtain a very thin and controlled Mo layer on the substrate, which is converted to MoS_2_ because of the exposure of the surface to S vapours. The final MoS_2_ layers are dependent on the thickness of the initial Mo layer, so it is important to have very precise control over the metal thickness and good homogeneity. MoO_3_ deposited by evaporation is an alternative approach to metallic Mo [71], since the oxide has a lower evaporation temperature and is easier to sulfurize.

The use of a Mo metal foil was suggested to obtain large flakes on 6-inch substrates [48]. The foil was placed 10 mm above a soda–lime glass substrate and using a mixture of Ar and O_2_ as carrier, flakes with later size higher than 400 μmm were obtained. Reactive MoO_2_ was produced thanks to the oxidation of Mo foil by the O_2_ carrier, and S powders were used as an S precursor. The dimensions of triangular MoS_2_ flakes were adjusted by controlling the distance between the Mo foil precursor and the substrate.

### 5.2. Sulfur

S powders are nontoxic and melt at about 115 °C, evaporating to gaseous sulfur. S vapour pressure is reported in [72] but the amount of precursors present in the vapour phase depends also on the quantity of powder placed in the crucible and on the their dimension: a good practice would be to replace the S in the crucible after each growth in order to increase reproducibility and have a better control of the deposited layers.

S powder with a weight in the range of 0.1–1 g is usually placed upstream in a quartz crucible about 10–15 cm away from the substrate. It is heated at temperatures of about 100–200 °C, and the vapour is delivered to the substrate by the carrier gas. In these conditions, the S vapour pressure is in the range of 10^−2^ – 1 torr. Since the evaporation of S is quite fast, a common technique to ensure repeatability is to independently heat the S crucible only after the substrate has reached the growth temperature setpoint. Moreover, in order to obtain a continuous supply of precursor during the whole process, the use of two different crucibles placed in different reactor position was suggested [73]. As already discussed above, changing the direction of the carrier gas to deliver the S vapours away from the substrate during the heating process may deliver more consistent results [34].

## 6. Liquid Precursors

The necessity to overcome the reproducibility and size limitations imposed by the use of powders fostered the research of alternative precursors. The idea of sulfurizing a thin and controlled Mo layer deposited on the substrate (originally obtained by sputtering or e-beam) was at the basis of the method of spinning liquid precursors. This approach, well known from the deposition of resists and protective layers, consists of dropping a viscous liquid with the Mo precursor onto the substrate and then using a fast-rotating spinner to spread and homogenize the liquid onto the surface. Once the centripetal and viscous forces are well balanced, the resulting layer has very good thickness reproducibility. Then, the substrate is placed in the reactor chamber and sulfurized using standard S powders.

### 6.1. Molybdenum

The literature has reported the use of two liquid precursors, sodium molybdate dihydrate (SMD—Na₂MoO₄·2H₂O) and ammonium molybdate tetrahydrate (AMT—(NH_4_)_6_Mo_7_O_24_). It is important to remember that SMD is incompatible with alkali metals, most common metals, and oxidizing agents. These precursors are usually diluted in water (0.1–0.2 g in 10–50 mL of H_2_O) and mixed with a promoter such as NaOH (0.1 g in 50 mL of H_2_O) and a density gradient medium such as iodixanol (Opti-Prep) to facilitate the spinning and substrate adhesion process. Standard proportions for these three components are in the range 1:(1–8):0.5, and they are then spun onto the substate for 30–60 s at 2000–4000 revolutions per minute. An hydrophilic surface is required to obtain a uniform coating molybdenum precursor, so the substrate is usually treated with O_2_ plasma and/or sodium hydroxide before the drop-casting of the solution [74].

SMD was also used in an alternative approach by embedding it into two thin pieces of glass that were fused together after heating. Thanks to the high temperature, the SMD melted and diffused through the molten glass to its surface with a dissolution–precipitation process. When the metal source reached the upper surface, it reacted with sulfur to grow MoS_2_ on the molten glass surface. Through this original method, highly uniform and monolayer MoS_2_ flakes on centimetre-scale glass substrates were obtained thanks to a uniform distribution of the metal precursor [75].

The liquid-phase precursor approach was used to obtain controlled doping of TMD and related heterostructures. Mixing of Mo liquid precursors along with selected dopants (Fe, Re, V) was reported to produce controllable doping in MoS_2_ with excellent uniformity. More complex structures, such as V-doped in-plane W_x_Mo_1−x_S_2_/Mo_x_W_1−x_S_2_, were also obtained [76]. Liquid, metal-organic-like precursors are still not common for MoS_2_ synthesis. Efforts are being directed towards developing suitable reagents to be used in MOCVD systems, in order to enhance process reproducibility and controllability. A notable effort was directed towards bis(tertbutylimido)bis(dimethylamido)molybdenum (Nt Bu)_2_(NMe2)_2_Mo, which was stored in a container kept at 45 °C and used as molybdenum precursor with H_2_S the as sulfurizing agent. A rapid and scalable process using a CVD approach on 2-inch sapphire substrate was developed to synthesize, layer-by-layer, atomically thin MoS_2_ films with good spatial homogeneity and wafer-scale uniformity [77].

### 6.2. Sulfur

Dimethyl disulfide (CH_3_SSCH_3_, DMDS) is an organic, flammable liquid with a garlic-like odour that can be used as a S precursor. It decomposes into methanethiol (CH_3_SH), ethene (CH_2_=CH_2_), carbon disulfide (CS_2_), and H_2_S. Reaction by-products of this precursor pose some safety hazards (see the following section) [78].

Dodecane-1-thiol (or dodecyl mercaptan, C_12_H_25_SH) is a sulfur-containing liquid thiol that can be bubbled and delivered into the reactor by using a carrier gas in order to obtain a precise and uniform concentration of S during the process [79].

## 7. Gaseous Precursors

### Sulfur

Despite its toxicity, hydrogen sulfide (H_2_S) has been used by some groups as a precursor for S. It decomposes very quickly at high temperature, and for this reason, it should be more efficient than S powder for TMD synthesis [80]. Its partial pressure during a typical TMD growth is in the range 10^−1^–10^−2^ [81,82]. H_2_S may also act as a reducing agent for metal oxides. It should be remembered that H_2_S is also corrosive, flammable, and explosive, with severe consequences in the event of exposure to eyes and skin and inhalation, and the design of a reactor with a H_2_S line should be carefully planned [83]. Despite these drawbacks due to safety reasons, a gas permit can be used to obtain a very controllable precursor supply, which is reflected in the reproducibility of the process.

## 8. Safety aspects of Mo and S Precursors

Ingestion of Mo or S related compounds is considered a very unlikely event in a research laboratory if standard precautions for the manipulation of substances are taken into consideration (e.g., wearing gloves, washing hands, etc.). However, it should be taken into consideration that a low level of Mo or S vapours could be present in the ambient because of the volatility of these substances at room temperature. Use of a laboratory hood should be always taken into consideration. Exposure to excess levels of Mo has been associated with adverse health outcomes such as respiratory effects following inhalation exposure to molybdenum trioxide, decreases in body weight, kidney damage, decreases in sperm count, and anaemia following oral exposure. A review of human and laboratory animal health effects for Mo can be found in [84]. Oral exposure to sulfur may cause problems in the gastrointestinal tract, central nervous system, kidneys, liver, heart, lungs, blood, and salivary glands. Sulfur has induced skin damage and affected various other tissues (https://www.bibra-information.co.uk/downloads/toxicity-profile-for-sulphur-1990/ Accesed on 12 September 2021). Handling of H_2_S exhaust could include a bubbler through an aqueous solution of ethylene glycol to dissolve H_2_S before releasing it to the atmosphere.

## 9. The Use of Growth Promoters in MoS_2_ Synthesis

The growth of 2D layers is driven by the chemical potentials and surface energies of the material and substrate. Depending on the balance between adhesion to the surface of atoms and adatom cohesive force, 2D layer growth might be favoured with respect to island nucleation. If the surface adhesion is insufficient to allow layer growth, the use of additional molecules (that are not involved in the chemical processes underlying the growth) is an attractive method to obtain 2D layers. These elements are generally known as “promoters”, and their use has been proven to be very beneficial for many aspects of the growth of MoS_2_.

The inclusion of other elements that may favour the nucleation and lateral growth of 2D layers was recognized as crucial in the very early steps of the research on the CVD growth of this material. In particular, the pioneering work of Ling et al. [85] analysed many possible options for promoters, giving an ample view of the possibilities; this was a fundamental paper to have a wide insight on many possible substances to be used as promoters. We limit our discussion to the most successful and useful promoters reported so far, to allow the reader a general view of the state of the art on this topic (Table 1). Many approaches have been proposed in these years, and we consider it useful to divide growth promoters into two main categories: inorganic solid state and organic. A section on the effect of surface treatments is also included, as this is another important element in determining the quality of the grown 2D layers.

From the point of view of understanding the mechanisms underlying the effects of promoters, theoretical work is currently underway, with different proposals to be investigated. For example, in [86], the effect of alkali metal compounds as promoters was studied, and the authors provided an explanation based on eutectic intermediates containing alkali metal molybdates and molybdenum oxides. Because of their low melting point, their mobility was enhanced in comparison with that of other molybdenum compounds, reducing the nucleation of new nanoislands and favouring lateral growth of existing ones.

**Table 1 materials-14-07590-t001:** Growth promoters for the synthesis of MoS_2_.

Seeding Promoter	Method	Substrate/Surface	Growth Temperature (°C)	Reference
*Organic*				
PTAS	dispersed on different substrate	SiO_2_/Si	>600	[85]
PTAS	dispersed as solution on growth substrate	SiO_2_/Si, quartz, sapphire, TiO_2_	650–680	[87,88,89]
PTAS	solution drop-casted on oxone-treated surface	graphene	-	[54]
PTAS	solution drop-casted on surface at 90 °C	SiO_2_/Si	750	[90]
PTARG	solution drop-casted on surface at 90 °C	SiO_2_/Si	750	[90]
F_16_CuPc	prior thermal evaporation on the growth substrate	SiO_2_/Si	650	[85]
PTCDA	solution drop-casted on surface	SiO_2_/Si, sapphire	650	[91]
PTCDA	solution dispersed on different substrate downstream	SiO_2_/Si	750	[92]
rGO	hydrazine solution drop-casted on surface	SiO_2_/Si, sapphire	650	[91]
CuPc	solution drop-casted on surface	SiO_2_/Si	680	[85,88]
CV (crystal violet)	solution drop-casted on surface	SiO_2_/Si	680	[88,93]
H2TPP (porphyrin)	thermal evaporation of thin film	SiO_2_/Si coated with carbon nanotubes	-	[94]
p-THPP	fibres dipped in promoter solution	graphene oxide fibres	650	[95]
Zn(II)-THPP	metalation of pTHPP promoter	graphene oxide fibres	650	[95]
** *Inorganic* **				
NaOH	added in liquid precursor solution	SiO_2_/Si	780	[96]
NaCl	on substrate facing the growth substrate	SiO_2_/Si	750	[97]
NaCl	mixed with solid Mo precursor	SiO_2_/Si, sapphire, Si, fused quartz, mica	650	[50]
Alkali metal halides (NaCl, KI)	placed upstream to growth substrate	sapphire	800	[98]
IIa metal chlorides (CaCl2, SrCl2)	spin-coating of solution on substrate	SiO_2_/Si, sapphire	850	[35]
Gold	EBL-patterned arrays of Au nanoparticles	SiO_2_/Si	650	[62]
Gold	drop-casting on colloidal Au nanoparticles	SiO_2_/Si	785	[63]

### 9.1. Inorganic Solid State Promoters

Among the promoters in this category, sodium was the first and most studied, as its beneficial effects on the lateral growth of flakes were recognized very early. The presence of Na on the substrate acts as a promoter for MoS_2_ flake nucleation. Moreover, the presence of Na facilitates the peeling of flakes, because it has a low energy barrier between MoS_2_ and the substrate during the growth process. This permits covering the flakes with a PMMA layer and then detaching the PMMA with the flakes with a simple immersion in water [48]. Na has been used in solution as sodium hydroxide (NaOH), particularly when liquid precursors were used, as it can be easily added to the solution to be spun on the substrate [96].

Another substance containing sodium that has shown usefulness in growing 2D materials is sodium chloride (NaCl). In particular, its use allowed obtaining high-quality MoS_2_–WS_2_ in-plane heterostructures [97]. NaCl’s role is related to the enhanced formation of micrometre-sized particles at lower temperature and to the weakening of interlayer adhesion, allowing for the lateral growth of materials. Different processes have been proposed for the role of NaCl, such as: (i) the formation of low-melting-point intermediates, which facilitates a sufficient supply of metal precursors; (ii) the decrease in the energy barrier for atom bonding at the edges of 2D flakes; and (iii) the generation of low-melting point-eutectic intermediates, increasing the surface mobility of precursors [35]. The beneficial effect of mixing NaCl into the MoO_3_ solid precursors was studied in [50], where 2D flakes were deposited on a variety of substrates. The authors indicated that the inclusion of NaCl allowed for a reduction in the growth temperature from 900 to 650 °C, making it a cost-effective growth promoter for high-quality, large-area monolayer flakes. In [99], the use of this salt was instrumental in obtaining large-area 2D layers of a large series of metal chalcogenides, including MoS_2_. The authors stated that NaCl helped in reducing the melting points of metal precursors and in increasing the chemical reaction rate to grow 2D TMCs.

In [98], alkali metal halides such as KI and NaCl were considered, highlighting their beneficial effect to the growth of MoS_2_ nanoflakes, which was due to the suppression of the nucleation of new MoS_2_ domains during growth and during subsequent enhancement of lateral growth of existing domains. In [100], the role of salts in promoting the 2D growth was interpreted as due to: (i) reaction between alkali metal salts with transition metal precursors forming nonvolatile liquid alkali metal molybdates that reduced the nucleation density and promoted lateral growth; (ii) catalytic effects that increased the surface reaction rate; (iii) the formation of highly active volatile metal oxychlorides possessing relatively low evaporation temperatures; and (iv) reduction in the activation energy on the specific surfaces of nonlayered materials hindering perpendicular growth.

Recently, the use of some group IIA metal chlorides (CaCl_2_ and SrCl_2_) as promoters was proposed [35], as it resulted in a relevant increase in the size of flakes, as shown in Figure 4. As in other cases, this promoter was spin-coated on the surface of the substrate as an aqueous solution, while the Mo and S precursors were in solid state. In this comprehensive work, different metal elements (Li, Na, Sr, Mg, Ca) were considered and an acid–base model was developed, allowing the authors to attribute the promoting effect to acid–base interactions between precursors and substrates influencing the adsorption of atoms for the formation of the 2D material.

Beneficial effects are not limited to these substances, since it has been reported that metals in solid state can also favourably influence the 2D growth of MoS_2_, as discussed above when the use of patterned substrates was introduced. More recently, it was reported that an increase in size of MoS_2_ flakes occurred after drop-casting a solution with colloidal gold nanoparticles [63]. This effect was explained by gold nanoparticles acting as catalytic seeding points for the initial synthesis of the MoS_2_ monolayer, similarly to what happens in 1D nanowires [101]. This is a topic of very relevant interest, as the integration of metal nanoparticles (such as gold or silver) with 2D nanostructures is very actively researched because of their effectiveness in enhancing and controlling the light-emission properties of these novel materials, allowing for in-depth studies of light–matter interaction in 2D systems [102,103,104].

### 9.2. Organic Promoters

Among the most-used organic promoters, perylene-3,4,9,10-tetracarboxylic acid tetrapotassium salt (PTAS) and perylene-3,4,9,10-tetracarboxylic dianhydride (PTCDA) have provided the most interesting results, thanks to the property of aromatic molecules of increasing the wettability of the growth surface and of lowering the free energy for nucleation. The first report on the usefulness of treating surfaces with such substances in obtaining large-area, high-quality TMD materials dates to 2012, when Y.H. Lee et al. used PTAS, PTCDA, and reduced graphene oxide (rGO) [91].

PTAS can be obtained by alkaline hydrolysis of PTCDA by diluting it in ethanol and adding KOH aqueous solution while refluxing. Ethyl ether can be added dropwise to the solution until a solid product begins to separate out. The precipitate can then be filtered and dissolved in deionized water, and the obtained aqueous solution must be filtered again to remove any insoluble residuals. Adding ethanol to the resulting aqueous filtrate allows precipitating the PTAS final product [105].

In a later study, Ling et al. [85] showed that PTAS permitted the growth of good-quality MoS_2_ monolayers at 650 °C, while the use of PTCDA allowed the use of lower temperatures while maintaining good quality in the nanostructures. As illustrated in Figure 5, when the promoter is deposited on a different substrate with respect of the growth one, the diffusion of the seeding molecules causes a concentration gradient from the promoter substrate over to the growth region.

The use of PTAS was recently demonstrated as very effective in achieving lateral heterostructures wherein MoS_2_ and other 2D materials can be parallel stitched [106]. Ref. [89] provided the possible growth mechanisms underlying the growth of nanoflakes, showing that this 2D material could be obtained on different surfaces such as Si particles, TiO_2_ aggregates, and quartz.

In [92], the effect of gradient of PTCDA on the growth dynamics was clarified, highlighting that a low amount of this organic promoter led to reduced lateral growth and boosted vertical growth, while a high amount enhanced lateral growth and suppressed vertical growth. Figure 6 illustrates the proposed growth dynamics due to the two opposite flows of precursors and PCTDA. This configuration allowed studying the effect of the gradient of this organic promoter on the MoS_2_ structure morphology and flake sizes along the length of the growth substrate.

As was shown recently by Martella et al. [91], the use of organic promoters can also be useful for tailoring the properties of 2D structures. Using two different perylene-based molecules (PTAS and perylene-3,4,9,10-tetracarboxylic dianhydride—PTARG) as seeding promoters resulted in a change in the local electronic polarizability of MoS_2_ monolayers due to extra charges trapped in the MoS_2_ monolayers.

Copper phthalocyanine (CuPc) and copper(II)-hexadecafluoro-29H,31H-phthalocyanine (F_16_CuPc) have also given interesting results [85,88], with performance comparable to PTAS and PTCDA in facilitating the growth of large-area, high-quality, and uniform monolayer and few-layer flakes. F_16_CuPc has the attractive property of high stability at high temperature, making this a good choice for high-temperature growth processes.

Another interesting organic compound is crystal violet (CV), thanks to the possibility of engineering its configuration by simply varying the polarity of solvent of the solution. Ko et al. [93] studied the effects of different CV conditions on the growth of MoS_2_ nanoflakes both from an experimental and a theoretical perspective. Their results gave interesting insight on the growth of monolayer MoS_2_ from aromatic seeding promoters.

Thin films of 5, 10, 15, 20-tetraphenylporphyrin (H2TPP) were used as a promoter layer for the realization of hybrid structures composed of carbon nanotubes and MoS_2_ nanosheets for the realization of flexible chemical sensors [94] and for the growth of vertical MoS_2_ nanosheets [107]. The use of this promoter also allows for the simultaneous doping of the material by performing a metalation of the H2TPP films, obtaining different metalloporphyrins such as Al(III)-tetraphenyl porphyrin (Al(III)TPP) or Zn(II) meso-tetra(4-hydroxyphenyl) porphyrin (Zn(II)THPP) [108]. Porphyrin-based organic molecules such as 5,10,15,20-tetrakis(4-hydroxyphenyl)-21H,23H-porphyrin (p-THPP) were used also for the growth of MoS_2_ on graphene oxide fibres. In [95], these fibres were dipped in the promoter solution. The use of Zn(II)THPP to achieve zinc-doping of the flakes was also demonstrated.

### 9.3. Surface Treatments

In their pioneering work, van der Zande et al. [26] showed that a careful treatment of SiO_2_/Si substrates (2 h in H_2_SO_4_/H_2_O_2_ (3:1) followed by 5 min of O_2_ plasma) and a minimal exposure of the precursors to air during storage was sufficient to obtain large flakes up to 100 µm. From these experimental observations, it is evident that the state of the surface on which the growth is carried on is of paramount relevance for the quality of flakes. From this point of view, many approaches to optimize this surface have been studied in the past. For example, the treatment of SiO_2_/Si substrates with oxygen plasma resulted in layer-controlled and large-area CVD MoS_2_ films [109].

Another interesting approach was studied recently that relied on the treatment of the SiO_2_ substrate by a piranha etching solution prior to deposition. The authors stated that this treatment reduced the surface free energy of the substrate, making the use of promoters unnecessary [110]. An interesting method to treat surfaces to ease the transfer of flakes was proposed by Shinde et al. [111]. By etching an SiO_2_ surface with hydrofluoric acid, the hydrophilicity is increased, allowing water to penetrate at the interface between polymer-capped MoS_2_ and the substrate, resulting in direct transfer of flakes over a large area.

## 10. Growth Mechanisms

The synthesis of MoS_2_ by CVD is the result of several steps, which can be summarized as follows: (i) transport of precursors to the substrate by a carrier gas; (ii) diffusion through the boundary layer from the gas phase to the substrate surface; (iii) adsorption of molecules to the substrate surface; (iv) diffusion of adsorbed molecules on the surface; (v) heterogeneous reactions on the surface resulting in TMD growth; (vi) desorption of by-products. In order to control the process and obtain a monolayer or few-layer material as an outcome, a high degree of control of the heterogeneous reactions should be pursued [35]. The exact evolution and progress of the synthesis and its mechanisms depend, of course, on the chosen precursors. The “holy grail” of MoS_2_ CVD would be to develop a precise, controllable, and reproducible process to uniformly deposit large-area monodimensional flakes over a large area without grain boundaries and with a low density of defects. Understanding the growth mechanisms for the different available precursors and considering the possible presence of growth promoters could help to recognize the strengths and weaknesses of these precursors.

Considering the experience gathered on graphene, but at the same keeping in mind that the synthesis of MoS_2_ proceeds through completely different mechanisms, in order to promote the growth of large 2D flakes, it is important to decrease the density of the nuclei and to increase the lateral growth rate while at the same time suppressing 3D nucleation [112]. Nucleation of MoO_3−x_ species at the beginning of growth is the first step to allow the formation of MoS_2_ nanoparticles and large flakes in the later stage of the synthesis [113]. It was suggested that this growth mode is essentially controlled by the deposition rate, which should be kept below a certain threshold to avoid the formation of thicker islands. Kang et al. [32] observed that this occurred when a low partial pressure of Mo vapour was used. With a higher Mo partial pressure, the layer-by-layer growth mode was abandoned in favour of the nucleation of a mixture of monolayer, multilayer, and no-growth regions. To obtain uniform monolayer growth, potentially on a large substrate, a precise and constant Mo partial pressure for the entire duration of growth is necessary. This requirement is not straightforward when Mo powders are used as precursors, since they deplete and are consumed as the growth proceeds, but is more easily obtained using metal-organic-like precursors such as Mo(CO)_6_.

The competition between the precursor mass flow towards the substrate and the reactions occurring at the surface determines the ultimate evolution of the growth. Zhou et al. [114] presented a general growth model for the deposition of a wide variety of TMDs, combining 12 transition metals and 3 chalcogens to synthesize up to 47 different 2D materials. They suggested that the mass flow supply determines the amount of metal precursors involved in the formation of the nucleus and the growth of domains, while the growth rate, essentially determined by the temperature, dominates the grain size of the layer. The interplay between growth rate and mass flow determines the formation of continuous monolayers, small flakes, atomic clusters, or large 2D single crystals. The two contradictory statements presented in [32,114] may be interpreted as a sign that the deposition of MoS_2_ with powders is very dependent on the used growth system and setup, and finding a unifying explanation of the experimental observation is not straightforward.

The main reason for the lateral enlargement of MoS_2_ flakes is the high chemical reactivity of the atoms at the edge of the flakes and in the presence of preferred nucleation sites, with high chemical reactivity, compared to the absence of dangling bonds in the vertical direction. A low and constant Mo precursor partial pressure may help in growing uniform monolayer thin films over a large area [32]. However, the growth of vertically standing MoS_2_ nanosheets was also reported [115]. They were supposed to be caused by the reduction in the elastic strain energy that forms during the horizontal growth of MoS_2_ nanosheets. The transition from 2D to 3D structure was achieved by controlling the quantity and distribution of the precursor concentration, placing the substrate in different orientations and positions with respect to the source powders [107]

When using Mo powders, the proposed reactions for MoS_2_ synthesis are:2 MoO_3_ + S → 2 MoO_3−x_ + SO_2_
2 MoO_3−x_ + (7 − x) S → 2 MoS_2_ + (3 − x) SO_2_

MoO_3_ is first reduced to suboxides (MoO_3−x_) by S vapours, further reduced to oxisulfides, and then transported to the substrate, where a further reaction with S vapour to form MoS_2_ occurs [116]. Since a MoO_3_ suboxide is involved in the growth reaction, the use of MoO_2_ instead of MoO_3_ was proposed in order to obtain better control of the flakes [117]. In this case, the synthesis reaction would proceed by the mechanism:MoO_2_ + 3 S → MoS_2_ + O_2_

This reaction avoids intermediate chemistry and permits obtaining higher-quality flakes on different substrates such as SiO_2_, Si, quartz, and SiN. The quantity of starting Mo powders greatly influences the reaction process. In [118], a comprehensive summary of typical powder weights, along with growth temperature and setup, was presented.

In order to have a detailed outlook on the growth process of MoS_2_, Cain et al. [119] deposited MoS_2_ using S and MoO_3_ powders on ultrathin SiO_2_ membranes to allow investigation of the nucleation mechanism with aberration-corrected STEM and elemental EDS mapping. They confirmed the presence of nucleation centres (10–30 nm) only at the centres of triangular flakes, representing the early stages of TMD growth. The second or third layer of the flakes also started at this site. Moreover, the nuclei showed a nanoscale core–shell structure, similar to that of inorganic TMD fullerenes, with inhomogeneous distribution of Mo and S. The growth process was essentially defined by the concentration of the chalcogen vapour: if the atmosphere was weakly reducing (sulfur poor), oxi-chalcogenide particles with an orthorhombic crystal structure were formed. In a moderately reducing sulfur-rich atmosphere, MoS_2_ formation was favoured. These experimental observations confirm the importance of promoting the formation of nucleation sites.

A convenient and empirical method to tune growth parameters towards the optimal growth conditions is to observe the flake shape at the optical microscope. Yang et al. [120] studied the morphological evolution of MoS_2_ flakes for various Mo–S ratios, using MoO_3_ and S powders as reagents at different temperatures. Obtaining a continuous film with large triangular MoS_2_ flakes should be possible at higher temperature and with a high S content (Figure 7).

Obtaining large flakes, by preventing the cohesion of smaller flakes, would also limit the presence of grain boundaries at flake coalescence regions, which may limit electrical transport in the layer [16]. Moreover, it was observed that the MoS_2_ bandgap could be tuned by controlling the distance from a grain boundary [121], so it is important to achieve good control over the flakes’ size over the substrate area in order to avoid undesired modulations of the bandgap. Nevertheless, it was demonstrated that MoS_2_ grain boundaries could exhibit a memristor behaviour [122], which may open new applications for this material.

An important source of carrier scattering in devices is defects and charged particles at the interface (Coulomb scattering). These defects are usually introduced in postprocessing fabrication steps, and it was suggested that proper passivation with a suitable dielectric could help mitigating the issue, increasing the performance of the device [123].

In general, point defects in MoS_2_ act as important electron scattering centres and may be categorized as vacancies or antisite defects [124,125]. Sulfur vacancies behave as deep donors and induce midgap defect states, making the material n-type. Furthermore, vacancies and antisites induce a modification in the electrical properties of MoS_2_, resulting in n-type doping and introducing localized states in the band gap, lowering the carrier mobility. They can cause Fermi level pinning and high contact resistance in electrical devices. Inclusion of other elements related to promoters (Na, Au) can also cause unintentional doping effects. These states also reduce the emission efficiency for photonic devices based on this material.

On the other hand, control of defect density and nature can be an effective tool to engineer material properties. It was shown that the electrical properties of MoS_2_ could be changed by exposing the samples to oxygen plasma, going from the semiconducting to the insulating regime [50]. A recent method to reduce sulfur vacancies was introduced by Durairaj et al. [126]; in this study, a SiO_2_/Si wafer was placed next to the growth substrate, providing oxygen and effectively enabling oxygen passivation of sulfur-vacancy defects in monolayer flakes. The authors observed a threefold increase in the PL efficiency due to the elimination of defect-related bound exciton emission.

In Figure 8, the reaction pathways that lead to the formation of MoS_2_ through intermediate species are summarized [118,127].

If MoCl_5_ is used instead of MoO_3_ powders, the proposed reactions are different from those reported above. Since the ratio between S and MoCl_5_ is very high (>1000), the conversion of MoCl_5_ to MoS_2_ is supposed to be completed in the gas phase, with a negligible concentration of intermediate suboxides. By precisely controlling the amounts of MoCl_5_ and S powders, their flow in the growth environment, and the total pressure in the growth chamber, Yu et al. [69] were able to control the number of MoS_2_ layers, with good uniformity over an area of several cm^2^. The key role was suggested to be the self-limiting nature of the layer-by-layer process, which could be controlled by tuning the flow of the precursors as well as the total pressure of the system: increasing the amount of precursors (from 1 mg of MoCl_5_ to 25 mg) and the total pressure (from 2 to 750 torr) resulted in the deposition of thicker films.

One could argue that the use of powder Mo precursors, because of their time-dependent evaporation and the necessity of precise and continuous flow control, would lead to worse and less reproducible results with respect to the sulfurization of a thin Mo layer predeposited on the substrate [118]. In the latter case, the resulting MoS_2_ layer would depend only on the starting conditions, which in principle could be precisely optimized by thermal evaporation or sputtering of the thin Mo or MoO_x_ layer. However, starting with a Mo film deposited on the substrate, the sulfurization process should occur at very high temperature to be efficient. This results in undesired surface evaporation of the starting material and leads to inhomogeneities, since the balance between the surface mobility and surface evaporation leads to limited grain sizes and/or limited yields of monolayer films. Moreover, it is still difficult to obtain an optimal starting film density and uniformity. However, one of the advantages of this approach is the ease of saturating the growth chamber with sulfur vapours (either from powders or from H_2_S), so this is not the process-limiting factor.

To promote bilayer growth, the interactions between the adatoms and the surface should be larger than the adatom–adatom interactions. If these conditions are not met, the deposition evolves towards a 3D growth. It was suggested that by increasing the intermolecular acid–base interactions between admolecules and the substrate surface, it should be possible to increase the adsorption. This can be done by selecting precursors and substrates with complementary acid/base properties [35]. A convenient way to increase the surface forces is to use growth promoters or seed molecules to change the surface energy, reduce the free energy barrier, and lower the nucleation energy necessary to obtain monolayer MoS_2_. It was argued that the best promoters are aromatic or graphene-like molecules that enhance the wettability of the substrate, lowering the free energy for the nucleation. In this way, better control of nucleation can be achieved [91]. Thus, an optimal concentration of the seeding promoter is essential for MoS_2_ monolayer deposition [128]. One important factor to be considered is the polarity of the growth promoter molecule, as pointed out by Ko et al. [93]. With density function theory calculations coupled to experimental evidence, they suggested that the polar part of the crystal violet molecule used as a promoter provides a preferred site for S adsorption, initiating the MoS_2_ nucleation. The orientation of the molecule during the spinning on the substrate, mediated by appropriate solvents, can effectively promote monolayer growth instead of multilayer MoS_2_ deposition.

It has been shown that the use of alkali-based promoters allows for larger flakes [98], but their excess might result in loss of MoS_2_ crystallinity, with the creation of additional defects [128]. Extensive work is currently underway on controlling defects in MoS_2_ flakes acting on growth parameters, not only to reduce their deleterious effects on devices, but also to take advantage of some of their features. For example, Na cations coming from promoters can cure interface defects to achieve low intrinsic defect levels and enhance electrical properties [129]. The use of liquid precursors spun in a controllable way on the substrate could overcome the problems due to the limited supply control of powder precursors. Spinning SMD and/or AMT with a density gradient medium and a growth promoter such as NaOH or NaCl could permit obtaining a reproducible layer that could be sulfurized in a controllable way. Kang et al. [130] presented a comprehensive study on the role of the different growth parameters in the synthesis of MoS_2_ flakes using liquid precursor, including the roles of temperature, S pressure, gas carrier flow, composition of the initial spun solution, and time. By an optimization of the process, MoS_2_ flakes with dimensions up to 100 μm were obtained in a reproducible way. AFM analysis of the MoS_2_ surface revealed the presence of precursor and promoter residues, particularly at the interfaces between different flakes. AMT is converted to MoO_3_ at about 300 °C. Using Na as a promoter, if further reacts with NaOH to produce sodium molybdate (Na_6_MoO_4_), which finally converts to MoS_2_ thanks to the presence of S vapour. Using SiO_2_ as a substrate, in this case, permits the reaction between Na_2_MoO_4_ and the oxide substrate to form sodium silicon oxide, promoting the lateral growth of the MoS_2_ layer [97].

## 11. Conclusions and Outlook

The rising interest in 2D MoS_2_ comes from its peculiar physical, chemical, and mechanical properties, including, among other features, peculiar light adsorption characteristics and promising electrical transport properties.

However, the synthesis of reproducible, uniform, and single-layer MoS_2_ flakes on large-area wafers is still a challenge, and open issues remain to be solved. Increasing control of the deposition process is mandatory to fine-tune the electrical and optical properties of MoS_2_. At the same time, obtaining flakes with larger dimensions, higher density, and a low concentration of defects is necessary to pave the way for the industrial application of this material. In this review, we explore the CVD synthesis of mono- and few-layer MoS_2_, giving an overview of the growth method, different substrates for deposition, and the used precursors and growth promoters. The topic of the doping of this material is also considered, and different growth mechanisms involving different types of precursors are overviewed.

From the critical analysis of the results obtained so far, it emerged that homogenous precursor feed, and in particular the achievement of fine control of Molybdenum supply during the CVD process, are necessary conditions for the optimization of the process. Because of their ease of use, the most used precursors for MoS_2_ synthesis are still Mo and S powders, but this approach presents several issues, such as poor control of the precursor supply causing low reproducibility. The use of growth promoters has allowed overcoming these problems, but the process is far from being optimized for an industrial scale-up. A more recent approach involves the use of liquid precursor spun on the substrate to obtain a controllable thin layer containing Mo and Na as promoter. This method permitted overcoming the problems faced by the other techniques to increase the reproducibility and yield of the process, resulting in very large flakes on very wide areas.

The vast majority of the CVD reactors used for MoS_2_ deposition have a horizontal geometry, which is known to have intrinsic flow inhomogeneities due to precursor depletion. This may be among the causes of the generally low reproducibility often observed in standard setups. Switching to a vertical gas distribution may help to solve these issues and obtain a better run-to-run consistency.

It is well known that CVD protocols are not easily transferrable from one laboratory to another, as even minimal changes in reactor parameters and growth conditions strongly affect reproducibility of the growth of 2D materials. It seems that many processes and recipes presented in literature worked only in a specific growth setup and configuration. There is a need to identify universal conditions and reaction schemes that will permit better control of the deposition of 2D MoS_2_ flakes. For this reason, further work on the theoretical modelling of growth is urgently needed to help identify the most crucial parameters. More experimental efforts focusing on the effects of other parameters somewhat overlooked (sulfur partial pressure, carrier gas flows, reactor geometry, and size) could also be useful to gain more knowledge.

From the point of view of reproducibility, the use of liquid precursors seems to have some advantages, as it allows a more precise and reproducible control of the amount of Mo source materials. Similarly, the use of sulfur powders is a crucial element, as the quantity of vapourized sulfur also depends on the mass of powder put in the crucible. The use of sulfur gas has shown some advantages, despite its safety drawbacks. Also, the precise control of gas flows is a fundamental parameter to achieve reproducibility of results.

In our opinion, high-quality, large flakes can be obtained in a controllable and reproducible way using a combination of these three fundamental elements: (i) liquid precursors, (ii) controlled surface treatments (etching and/or plasma), and (iii) organic promoters (Na-based ones being the most effective). However, the optimal combination of these elements has still not been clearly identified, and exact protocols and recipes are still very cumbersome; more research work is needed to identify the optimal conditions.

From the work presented in the literature, it seems that the uniformity of large flakes is strongly dependent on the state of the growing surface. Reduced graphene and sapphire seem to have an advantage, but by careful etching and cleaning of the substrate (by methods such as plasma cleaning), very good results can be obtained also on more common and less expensive systems such as SiO_2_/Si. These excellent results are fostering a large increase in studies and activities on the growth of this material, considering different substrates on which to grow it, various elements for doping, and the development of reliable models to gain insight on the growth mechanisms. Variations on the basic CVD concepts to achieve better control of precursor supply through various hardware modifications of the apparatus have been proposed with encouraging results.

From this perspective, the knowledge on the CVD growth of MoS_2_ seems to be mature enough to envision the design and realization of complex heterostructures based on MoS_2_, in which not only the materials but their dimensionality are different (so-called hybrid heterostructures).

However, we think that at the moment, there is a bottleneck on the realization of complex heterostructures, as the transfer of flakes from one substrate to another is a very complex and delicate step that could represent a second major hurdle for the transfer of the technology for this material to the development of commercial devices. Therefore, it would be desirable to achieve large-area flakes on other substrates, such as metals, semiconductors, and possibly flexible substrates, on which devices can be directly fabricated.

Moreover, we are keen to consider that the next major breakthrough for MoS_2_ is related to the possibility of realizing patterned deposition of this 2D material, aiming for a method to obtain designed microcircuits. From this point of view, the possibility of “printing” MoS_2_ layers seems to us a more appealing approach than delicate and cumbersome transfer and manipulation of large flakes.

## Figures and Tables

**Figure 1 materials-14-07590-f001:**
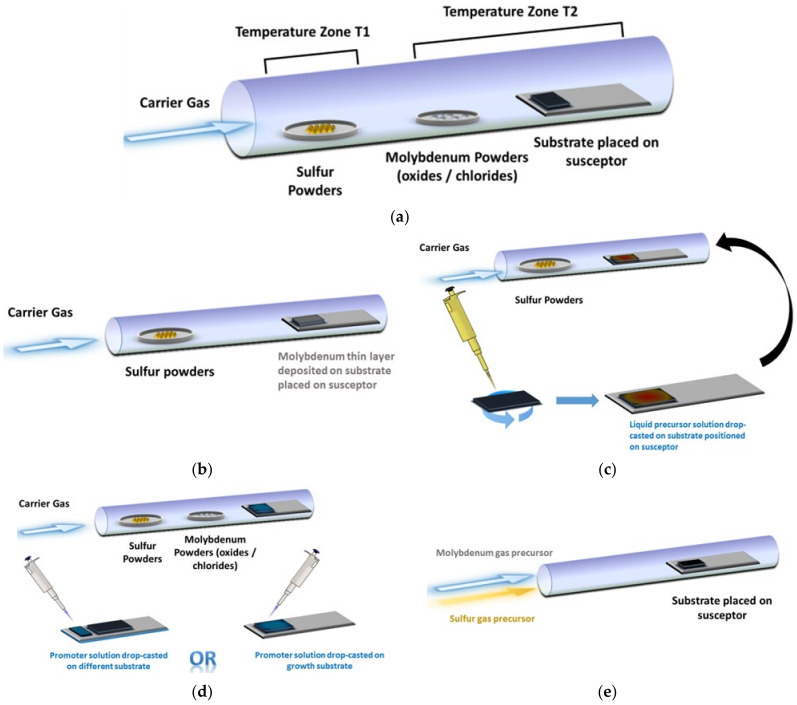
Schematic view of the CVD tube in different configurations for MoS_2_ flake growth: (**a**) with solid precursors separated by substrate; (**b**) with solid molybdenum deposited on growth substrate; (**c**) with liquid molybdenum precursors; (**d**) with solid precursors and drop-casted promoters (either on growth substrate or a different substrate); (**e**) with gaseous precursors.

**Figure 2 materials-14-07590-f002:**
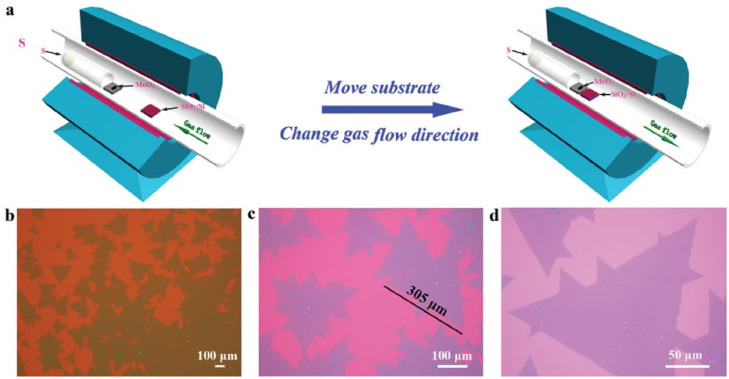
(**a**) Flow reversal method to improve MoS_2_ deposition; (**b**–**d**) typical optical images of MoS_2_ flakes obtained with the modified procedure. Adapted from [34].

**Figure 3 materials-14-07590-f003:**
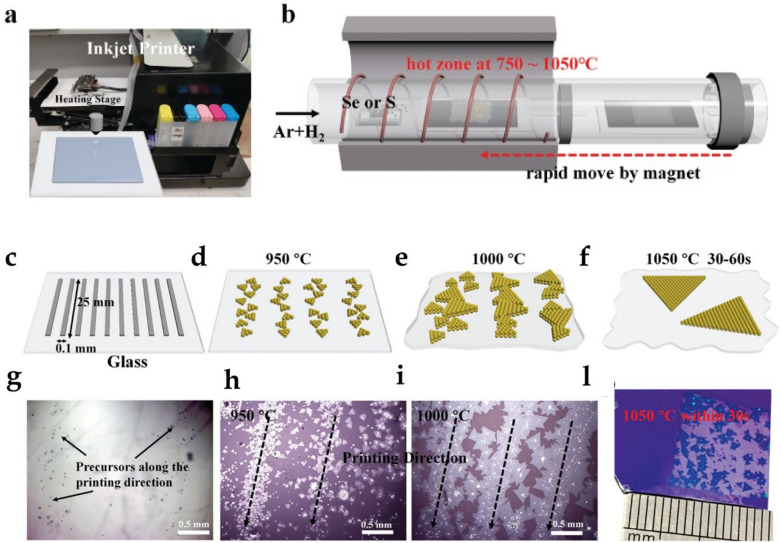
Synthesis of MoS_2_ flakes from inkjet-printed aqueous precursors: (**a**) a photo of the customized inkjet printer; (**b**) schematic of the synthesis process with the fast annealing process; (**c**–**f**) growth of flakes at different growth temperatures; (**g**–**l**) corresponding optical images. Adapted from [64]. Copyright 2021 John Wiley & Sons.

**Figure 4 materials-14-07590-f004:**
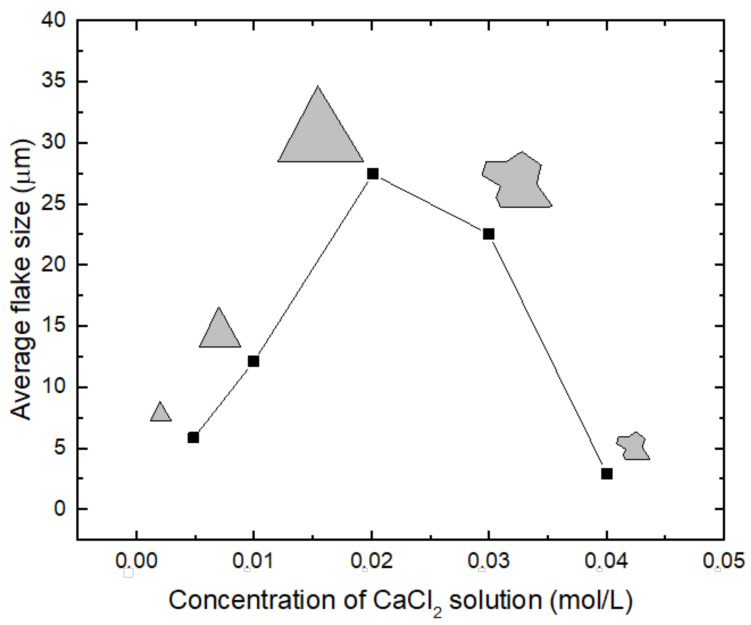
Dependence of the average size of flakes on CaCl_2_ solution concentration. Symbols indicate schematically the morphology of flakes.

**Figure 5 materials-14-07590-f005:**
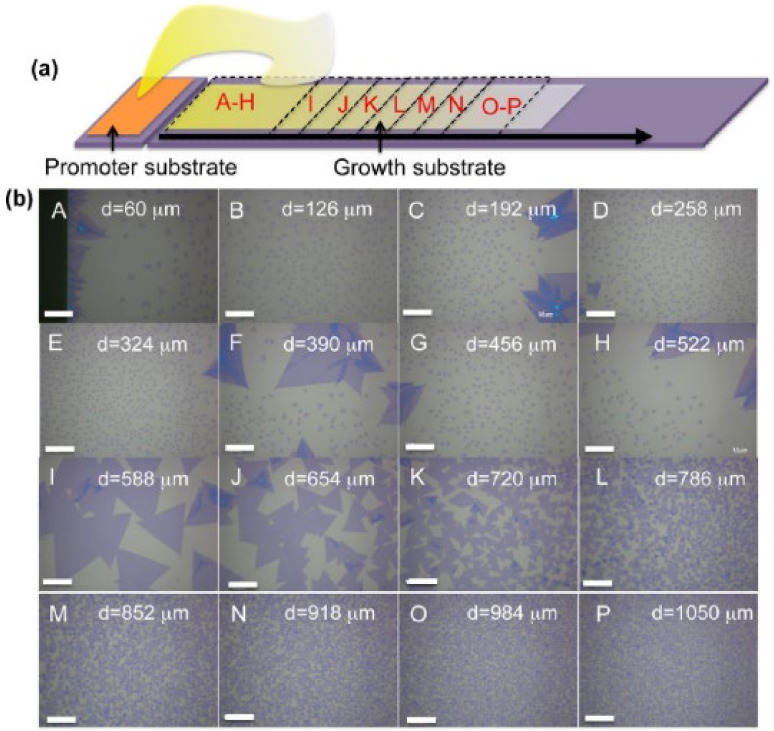
(**a**) Schematic of substrates used in PTAS-promoted growth of MoS_2_ flakes. (**b**) Optical images of structures in different regions of the growth substrate, as identified above. Scale bars: 20 μm. Adapted with permission from [85]. Copyright 2014 American Chemical Society.

**Figure 6 materials-14-07590-f006:**
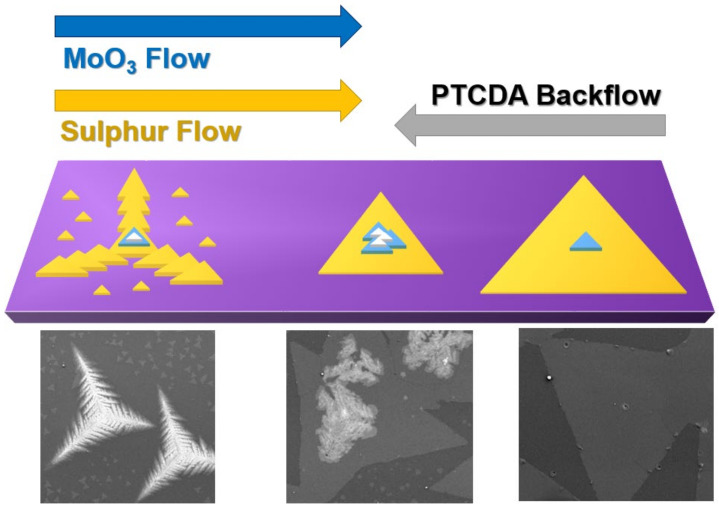
Illustration of the growth dynamics of MoS_2_ due to the flow gradient of PTCDA organic promoter, with SEM images from different areas of the substrate. Reprinted from [92].

**Figure 7 materials-14-07590-f007:**
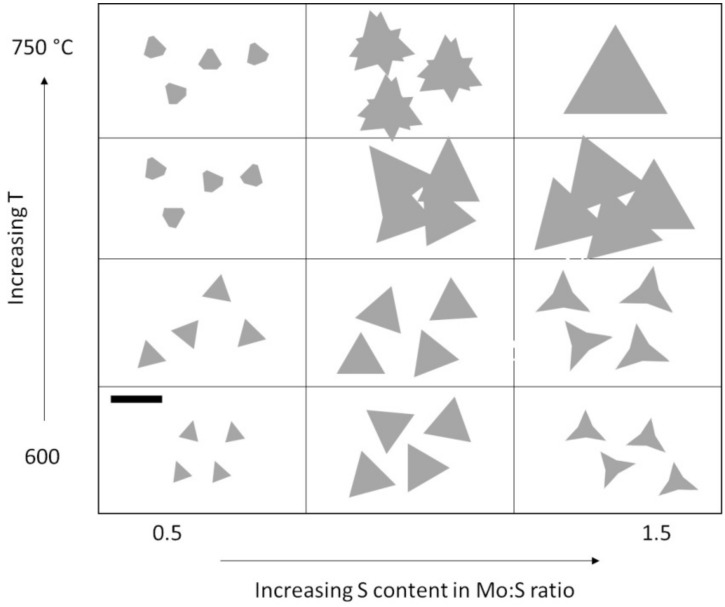
Schematic illustration of the evolution of the morphology of MoS_2_ flakes for changing Mo–S ratios and temperatures. Marker represents a flake size of 100 μm. Derived from data from [120].

**Figure 8 materials-14-07590-f008:**
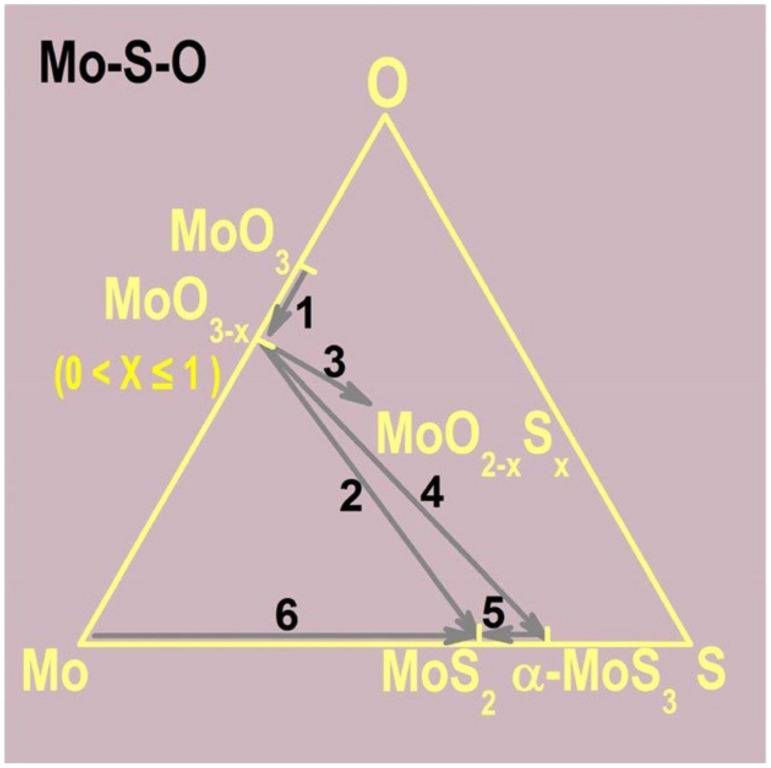
Ternary phase diagram for Mo-S-O, showing the possible reactions to produce MoS_2_. Adapted from [127]; reprinted with permission from AAAS.

## Data Availability

No new data were created or analyzed in this study.

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
