# Peer review of "A Review on Chemical Vapour Deposition of Two-Dimensional MoS_2_ Flakes"

_materials, 2021, doi:10.3390/ma14247590_

Round 1

Reviewer 1 Report

2D semiconductors have received much interest these years due to their excellent properties for future electronics and optoelectronics. CVD growth of 2D materials is a prerequisite for these applications. In this review, the authors systematically summarized recent advance in the CVD growth of MoS2, one of the most important 2D semiconductors.

Overall, the review is comprehensive, informative, and useful for readers in the field. However, several very relevant references are not mentioned in the current version which should be included in the revised manuscript. In addition, the depth of the review can be further improved. With these changes, the manuscript can be considered for publication in Materials as a decent work.

Our technical questions are below.

  • Fig 1a, the scheme is not exactly correct because molybdenum itself is rarely used as precursor for MoS2 growth. Usually, metal oxides or chlorides are used due to their high vapor pressure.

  • Figure quality needs to be improved.

  • The conclusion part should be further strengthened as the current version is too simple and general. For example, could you please put forward some possible methods which may solve the reproducibility/uniformity problem of the growth of 2D TMDCs? Any ideas on how to prepare large single crystals? How to guarantee the property uniformity of 2D MoS2?

  • Regarding the use of new types of sulfur precursor for CVD growth of 2D MoS2, gas phase and liquid phase sulfur precursors should be discussed. For example, Ref 1. Vertical Chemical Vapor Deposition Growth of Highly Uniform 2D Transition Metal Dichalcogenides. Acs Nano 2020, 14 (4), 4646-4653. Ref 2. Synthesis of Ultrahigh‐Quality Monolayer Molybdenum Disulfide through In Situ Defect Healing with Thiol Molecules. Small 2020, 16, 2003357.

  • Page 22, Line 814, the authors wrote “A more recent approach involves the use of liquid precursor, spinned on the substrate to obtain a controllable thin layer containing Mo and Na as promoter”. Reference should be provided here.

  • Page 22, Line 808, the authors wrote “it emerges that the achievement of a fine-control of Molybdenum supply during the CVD process is a necessary condition for the process optimization.” Regarding controlling the metal source, a recent new method is suggested to be discussed. Ref 1. Dissolution-Precipitation Growth of Uniform and Clean Two Dimensional Transition Metal Dichalcogenides. National Science Review 2021, 8, nwaa115.

  • Besides uniformity and reproducibility, material quality if another big problem in 2D MoS2. The authors are suggested to discuss this point. For example, how to reduce defects during growth or after growth? How these defects may affect device performance of 2D semiconductors?

Author Response

2D semiconductors have received much interest these years due to their excellent properties for future electronics and optoelectronics. CVD growth of 2D materials is a prerequisite for these applications. In this review, the authors systematically summarized recent advance in the CVD growth of MoS2, one of the most important 2D semiconductors.

Overall, the review is comprehensive, informative, and useful for readers in the field. However, several very relevant references are not mentioned in the current version which should be included in the revised manuscript. In addition, the depth of the review can be further improved. With these changes, the manuscript can be considered for publication in Materials as a decent work.

Our technical questions are below.

Fig 1a, the scheme is not exactly correct because molybdenum itself is rarely used as precursor for MoS2 growth. Usually, metal oxides or chlorides are used due to their high vapor pressure.

The Referee is right, we modified the text with “molybdenum powders” and “sulfur powders” for clarity. Moreover, we worked on the figure 1, adding some modifications to improve its quality: elements of the schematics have been better aligned, text has been added to improve clarity, the size of some figures have been increased. Also Figure 1d was substantially modified to avoid confusion between the two different options for drop-casting of the prometer solution.

Figure quality needs to be improved.

we have modified and improved figure 1 as indicated in the previous comment. Other figures are authorized reprints or adaptations of figures from other publications

The conclusion part should be further strengthened as the current version is too simple and general. For example, could you please put forward some possible methods which may solve the reproducibility/uniformity problem of the growth of 2D TMDCs? Any ideas on how to prepare large single crystals? How to guarantee the property uniformity of 2D MoS2?

We thank the referee for this comment and his invitation to improve the conclusions. We have added several paragraphs to the conclusion, addressing all issues raised.  In the conclusions we have also argued what is, in our opinion, a crucial bottleneck for the realization of complex heterostructures towards, that is the need of transfer flakes from the growth substrate.

Regarding the use of new types of sulfur precursor for CVD growth of 2D MoS2, gas phase and liquid phase sulfur precursors should be discussed. For example, Ref 1. Vertical Chemical Vapor Deposition Growth of Highly Uniform 2D Transition Metal Dichalcogenides. Acs Nano 2020, 14 (4), 4646-4653. Ref 2. Synthesis of Ultrahigh‐Quality Monolayer Molybdenum Disulfide through In Situ Defect Healing with Thiol Molecules. Small 2020, 16, 2003357.

We thank the reviewer for the suggestion. We added Dodecane-1-thiol as liquid precursor, as indicated by ref.2. The precursors used in ref.1 (H2S and liquid Mo precursor) were already present in the discussion, but ref. 1 was indeed useful for enhancing the discussion about the reactor geometry (vertical or horizontal) that was expanded in Page 3.

Page 22, Line 814, the authors wrote “A more recent approach involves the use of liquid precursor, spinned on the substrate to obtain a controllable thin layer containing Mo and Na as promoter”. Reference should be provided here.

The referee indicates a sentence of the conclusion, where we intentionally decided to not repeat the references indicated in the previous part of the review. In the conclusions we recap what we discussed in the whole paper, so we do not think references are needed again for each statement. The sentence for which the referee is asking references is discussed thoroughly in the “liquid precursor” paragraph and just before the conclusion (end of page 22- “Kang et al. [132] presented a comprehensive study on the role of the different growth parameters in the synthesis of MoS2 flakes using liquid precursor, including the role of temperature, S pressure, gas carrier flow, composition of the initial spinned solution and time”)

Page 22, Line 808, the authors wrote “it emerges that the achievement of a fine-control of Molybdenum supply during the CVD process is a necessary condition for the process optimization.” Regarding controlling the metal source, a recent new method is suggested to be discussed. Ref 1. Dissolution-Precipitation Growth of Uniform and Clean Two Dimensional Transition Metal Dichalcogenides. National Science Review 2021, 8, nwaa115.

We thank the referee for this interesting method. We added the suggested referee to the precursor section and discussed the proposed approach.

Besides uniformity and reproducibility, material quality if another big problem in 2D MoS2. The authors are suggested to discuss this point. For example, how to reduce defects during growth or after growth? How these defects may affect device performance of 2D semiconductors?

We thank the referee for this important suggestion on a topic that was overlooked by us in the first version of the manuscript. We added some discussion about this issue in the introduction with an important reference, and we further discuss this topic in the growth mechanism and conclusion sections

Reviewer 2 Report

Rejected
Comment: This review demonstrated the detailed chemical vapor deposition (CVD) synthesis 
procedure for MoS2 nanoflakes in several aspects including the technique details, precursors 
and promoters, as well as the corresponding doped MoS2 flakes. To be honest, CVD growth 
methods of MoS2 nanoflakes can be useful for researchers specializing in relevant fields 
although the strategies are indeed important. However, the details of CVD method of MoS2
and other transition metal dichalcogenides (TMDs) have ever been summarized in a lot of 
reviews so that I would like to know the differences of this review and the previous ones. In 
the present version, unfortunately, I cannot find some important and interesting opinions that 
are put forward by the authors. The authors need to think about the relationship between the 
methodology and crystal growth, and to demonstrate the underlying mechanism of CVD 
growth. Therefore, I recommend against the publication of the manuscript due to an amount 
of problems, but it can be considered after careful and complete revision. Some detailed 
comments are listed below.
1. The title should be considered carefully. The words “technique”, “precursors” and 
“promoters” are not parallel relation, because precursors and promoters may be parallel to 
the control of temperature or pressure but not to the CVD technique. That is, the sections of 
the review should also be revised.
2. The introduction should be revised for concision. MoS2 is almost not mentioned in the first 
three paragraphs (Line 27~55) I do know the importance of the background but it should be 
concise enough. In addition, the review theme is MoS2 so it is not appropriate to introduce all 
the TMDs in a general way (Line 62~80). Similarly, CVD method should be the key point of 
the review but it is inconspicuously mentioned with atomic layer deposition (ALD) in the 
introduction (Line 81~84). Finally, it is confusing to list the precursors in detail in introduction
(Line 85~92).
3. The statement, “However, techniques to obtain MoS2 flakes with large size (>50 μm) in 
reproducible ways are far from being assessed”, is not appropriate. Excellent results have been 
reported from different groups. Some reviews have been listed: Chem. Rev., 2018, 118, 6236–
6296; Chem. Rev., 2018, 118, 6134–6150; iScience, 2021, 24, 103229. Moreover, these 
reviews have demonstrated CVD growth of large-scale MoS2 single crystals and films in the 
aspects of methodology and mechanism. There is the question–what the new opinions of this 
review are.
4. The logic of the review is not clear enough. The section “Doping MoS2 layers” (Line 444) is 
not appropriate to be placed here although precursor design is essential to doping process. 
The concerns are not the same for obtaining doping materials and large-scale crystals (or 
films). The former one focuses on dopant selection and structural modulation process while 
the latter one aims at nuclei and growth processes. In my opinion, this part should be removed 
or placed just before the summary as an additive part, because doping can be easily achieved 
beyond CVD (such as plasma, ion beam irradiation).
5. The theoretical mechanism and analysis of different CVD methods should be added to the 
revised version. Excellent comments are useful and significant for readers rather than 
reference collection. For example, the mechanism of promoters have been expounded clearly 
in most articles but I cannot gain any useful information about theoretical mechanism or 
physical explanation in this review. 
6. A lot of writing and format mistakes should be revised carefully. I just point out some of 
them. 1) “two-dimensional” appears differently in title and abstract. 2) Reference numbers are 
written in a wrong format (Line 92). 3) The letters a, b, c, etc. are in different fonts (e.g. Fig. 1 
and Fig. 2). 4) Fig. 3 is divided into two parts in different pages. 5) The font is different (Line 
328–330). 6) The abbreviations in Line 388 are repeated with full words and abbreviations 
appearing in Line 89–90. 7) Section titles in Line 417 and 427 are different from the similar 
ones above

Author Response

Comment: This review demonstrated the detailed chemical vapor deposition (CVD) synthesis 
procedure for MoS2 nanoflakes in several aspects including the technique details, precursors 
and promoters, as well as the corresponding doped MoS2 flakes. To be honest, CVD growth 
methods of MoS2 nanoflakes can be useful for researchers specializing in relevant fields 
although the strategies are indeed important. However, the details of CVD method of MoS2
and other transition metal dichalcogenides (TMDs) have ever been summarized in a lot of 
reviews so that I would like to know the differences of this review and the previous ones. In 
the present version, unfortunately, I cannot find some important and interesting opinions that 
are put forward by the authors. The authors need to think about the relationship between the 
methodology and crystal growth, and to demonstrate the underlying mechanism of CVD 
growth. Therefore, I recommend against the publication of the manuscript due to an amount 
of problems, but it can be considered after careful and complete revision. Some detailed 
comments are listed below.

We thank the referee for the constructive comments and observations.

We carefully revised the manuscript adding some points in the introduction, the discussions and the conclusions concerning what are, in our opinion, the most important challenges for the growth of MoS2 by CVD (detailed below).

We agree that there are many reviews on transition metal dichalcogenides, but we also believe that our paper, being focused entirely on CVD of MoS2 flakes, may be indeed useful for researchers working in this particular field. Usually in other reviews only limited sections (1-2 pages) are dedicated to this topic, despite the fact that only in the last 2 years more than 400 papers have been published and 20K citations have been recorded on this growth method for this material (source: Web Of Science on “CVD + MoS2”). In these last two years many novel methods and improvements have been reported and we felt the need for a comprehensive review of the state-of-the-art on this specific topic.

We discussed the use of growth promoters and of the newly introduced liquid precursors, which in our opinion are one of the most interesting and promising approaches to solve some of the problems of the synthesis of this material. These topics, liquid precursors in particular, were rarely covered by other reviews. We also provided an overview and comments of the most recent literature, with very relevant breakthroughs that older reviews could not report about (among all, probably the most impactful is the possibility of using an inkjet printer to obtain patterned growth of flakes).

We also think that the possibility of growing large flakes on different substrates (already well demostrated, as we discuss in the dedicated section) should be attracting more interest, as this could allow to eliminate the delicate and crucial step of transfer of flakes. This is an important point for the technology transfer of this fabrication method towards mass production.

A second very important issue considers growth reproducibility, another necessary requirement for technology transfer. In our opinion theoretical and experimental work focusing on some overlooked parameters can allow to increase the reliability of the CVD protocols. Usually, the growth parameters and recipes proposed in literature only work on specific reactors and growth conditions, and there is an urgent need to identify and define a general framework for the growth of 2D MoS2 independent of the particular hardware.

We believe that with these changes the paper will be more interesting and useful, and we hope to have fulfilled the referee’s requests.

  1. The title should be considered carefully. The words “technique”, “precursors” and 
    “promoters” are not parallel relation, because precursors and promoters may be parallel to 
    the control of temperature or pressure but not to the CVD technique. That is, the sections of 
    the review should also be revised.

We changed the title to “A review on Chemical Vapour Deposition of 2-dimensional MoS2 flakes” to enhance the clarity of our message. We think our paper is arranged in a logical way, following the various steps of the MoS2 synthesis itself as it would be approached from a newcomer scientist: Hardware equipment (i.e. CVD setup), choice of the  substrate and of the precursors. Than, a discussion on growth promoters is added as they are a requirement to optimize the growth, while the discussion of the growth mechanisms concludes the paper.

.

  1. The introduction should be revised for concision. MoS2 is almost not mentioned in the first 
    three paragraphs (Line 27~55) I do know the importance of the background but it should be 
    concise enough. In addition, the review theme is MoS2 so it is not appropriate to introduce all 
    the TMDs in a general way (Line 62~80). Similarly, CVD method should be the key point of 
    the review but it is inconspicuously mentioned with atomic layer deposition (ALD) in the 
    introduction (Line 81~84). Finally, it is confusing to list the precursors in detail in introduction
    (Line 85~92).

The introduction was shortened and rearranged for clarity

  1. The statement, “However, techniques to obtain MoS2 flakes with large size (>50 μm) in 
    reproducible ways are far from being assessed”, is not appropriate. Excellent results have been 
    reported from different groups. Some reviews have been listed: Chem. Rev., 2018, 118, 6236–
    6296; Chem. Rev., 2018, 118, 6134–6150; iScience, 2021, 24, 103229. Moreover, these 
    reviews have demonstrated CVD growth of large-scale MoS2 single crystals and films in the 
    aspects of methodology and mechanism. There is the question–what the new opinions of this 
    review are.

The referee is right and we rewrote the sentence in page 3 (introduction), commenting the size of the flakes grown in literature and the issues to still be adreessed. We also added some of the references suggested. We also have expanded consistently the conclusions, adding our opinions on the most crucial issues still facing the growth of MoS2, despite the success in achieving large-area flakes.

  1. The logic of the review is not clear enough. The section “Doping MoS2 layers” (Line 444) is 
    not appropriate to be placed here although precursor design is essential to doping process. 
    The concerns are not the same for obtaining doping materials and large-scale crystals (or 
    films). The former one focuses on dopant selection and structural modulation process while 
    the latter one aims at nuclei and growth processes. In my opinion, this part should be removed 
    or placed just before the summary as an additive part, because doping can be easily achieved 
    beyond CVD (such as plasma, ion beam irradiation).

We followed the referee suggestions and moved the information about doping in the introduction, shortening and simplifying the discussion. Nevertheless, we think that this is an important topic, deserving a brief recap.

  1. The theoretical mechanism and analysis of different CVD methods should be added to the 
    revised version. Excellent comments are useful and significant for readers rather than 
    reference collection. For example, the mechanism of promoters have been expounded clearly 
    in most articles but I cannot gain any useful information about theoretical mechanism or 
    physical explanation in this review. 

We added several paragraphs to the discussion, following the observation of the reviewer. In particular, we highlighted the relevance of the nucleation mechanism for the growth of flakes and the fundamental role of the chalcogen vapour in driving the process.

  1. A lot of writing and format mistakes should be revised carefully. I just point out some of 
    them. 1) “two-dimensional” appears differently in title and abstract. 2) Reference numbers are 
    written in a wrong format (Line 92). 3) The letters a, b, c, etc. are in different fonts (e.g. Fig. 1 
    and Fig. 2). 4) Fig. 3 is divided into two parts in different pages. 5) The font is different (Line 
    328–330). 6) The abbreviations in Line 388 are repeated with full words and abbreviations 
    appearing in Line 89–90. 7) Section titles in Line 417 and 427 are different from the similar 
    ones above

We carefully revised the paper for typos and mistakes and we hope it is better now. We thank the referee for pointing out some errors, allowing us to improve the general quality of the manuscript.

Reviewer 3 Report

According to Manuscript ID: materials-1461006, entitled „A review on Chemical Vapour Deposition of 2-dimensional 2 MoS2 flakes: technique, precursors and promoters“ by authors Luca Seravalli and Matteo Bosi.

The paper presents a detailed review of CVD growth of 2 D layers of MoS2 flakes, describing the CVD reactors, precursors, surface treatmeants, substrates etc. The growth mechanisms are also proposed. The paper is well organized, written in good style and easy to read.

The paper is worth to be published in its present form.

Author Response

We thank the reviewer for the positive comments.

Round 2

Reviewer 2 Report

The problems I pointed out have ever been revised so I recommend the publication of the manuscript.